# Incremental Learning via Robust Parameter Posterior Fusion

## ABSTRACT

The posterior estimation of parameters based on Bayesian theory is a crucial technique in Incremental Learning (IL). The estimated posterior is typically utilized to impose loss regularization, which aligns the current training model parameters with the previously learned posterior to mitigate catastrophic forgetting, a major challenge in IL. However, this additional loss regularization can also impose detriment to the model learning, preventing it from reaching the true global optimum. To overcome this limitation, this paper introduces a novel Bayesian IL framework, **R**obust **P**arameter **P**osterior **F**usion (RP$^2$F). Unlike traditional methods, RP$^2$F directly estimates the parameter posterior for new data without introducing extra loss regularization, which allows the model to accommodate new knowledge more sufficiently. It then fuses this new posterior with the existing ones based on the Maximum A Posteriori (MAP) principle, ensuring effective knowledge sharing across tasks. Furthermore, RP$^2$F incorporates a common parameter-robustness priori to facilitate a seamless integration during posterior fusion. Comprehensive experiments on CIFAR-10, CIFAR-100, and Tiny-ImageNet datasets show that RP$^2$F not only effectively mitigates catastrophic forgetting but also achieves backward knowledge transfer.

## CCS CONCEPTS

• **Computing methodologies** → **Lifelong machine learning**; *Bayesian network models*; Image representations.

## KEYWORDS

Incremental Learning, Lifelong Learning, Catastrophic Forgetting, Bayesian Theory

**ACM Reference Format:**
. 2024. Incremental Learning via Robust Parameter Posterior Fusion. In *Proceedings of ACM MULTIMEDIA (MM)*. ACM, New York, NY, USA, 10 pages. https://doi.org/10.1145/nnnnnnn.nnnnnnn

## 1 INTRODUCTION

Acquiring knowledge continuously is essential for intelligent systems operating in ever-changing environments, a process referred to as Incremental Learning (IL) [12, 33]. In contrast to the conventional training paradigm that amasses all data at once, IL adopts a task-by-task training paradigm, where only a subset of data is available for each training task. Training within this setting, models encounter the formidable challenge of catastrophic forgetting [17, 35], wherein the assimilation of new knowledge often results in diminished performance on previously learned tasks.

**Unpublished working draft. Not for distribution.**

To mitigate catastrophic forgetting, a fundamental strategy is to identify and preserve the parameters critical to past tasks. This strategy has directed the development of numerous IL methodologies [1, 9, 34, 41, 68], including techniques grounded in Bayesian theory [20, 22, 32, 39, 46]. The Bayesian-based approaches utilize Laplace approximation to model the parameter posterior distribution of old tasks as a Gaussian distribution. This Gaussian posterior then serves as a priori to guide subsequent network training, enforcing regularization to prevent substantial deviations from this priori. In this context, the variance of the Gaussian distribution (i.e., the Hessian through Laplace approximation) acts as a parameter importance matrix to identify critical parameters.

While the regularization has proven its effectiveness in IL, controlling its strength poses a considerable challenge for Bayesian-based methods. A particular issue is ensuring that parameter adjustments through regularization accurately reflect the Maximum A Posteriori (MAP) estimate of the joint posterior distribution for both new and old tasks. Particularly, achieving a harmonious integration of the regularization (representing the old-task posterior) with the learning loss for new tasks remains a challenge [21], even when employing predefined weights. Furthermore, the necessity for accommodating new data within IL contexts inevitably compels the parameters to diverge from the old-task posterior, potentially leading to information loss.

To address the above challenges, this paper introduces a novel Bayesian IL framework, referred to as Robust Parameter Posterior Fusion (RP$^2$F). Unlike traditional methods that integrate the posterior of old tasks through regularization, RP$^2$F constructs a parameter posterior for the new task and then directly fuses it with the posterior from previous tasks. This integrated posterior enables the derivation of a closed-form MAP solution for parameters, facilitating a more coherent balance across tasks. As the learning process evolves, parameter estimates continuously adjust from the old-task MAP with the introduction of each new task, which may trigger potential forgetting. To mitigate this, RP$^2$F incorporates a parameter robustness priori within the training regimen, which bolsters the model's resilience against changes in parameters.

To implement RP$^2$F, two critical techniques are imperative for enhancing performance: the computation of the Hessian matrix in Laplace approximation and the design of the parameter robustness priori. Existing methods [22, 46] typically approximate the diagonal of the Hessian matrix with the Fisher information matrix due to the computational and storage constraints, where the Fisher information matrix only depends on the model's first-order derivative information. Instead, this paper develops a perturbation-based method for estimating the Hessian information, which leverages derivatives with respect to small perturbations applied to the parameters. Furthermore, our analysis reveals that a network's robustness is closely linked to the uniformity of the singular values of the extracted feature matrix. Drawing on this insight, we propose regularizing the features during training to balance the singular values, thus diminishing the potential forgetting issue caused by parameter changes.

In summary, the core contributions of this research are listed as follows:

- We propose a novel Robust Posterior Parameter Fusion ($RP^2F$) framework for IL, which leverages the Bayesian theory to derive a posterior fusion method across tasks.
- To facilitate Laplace posterior estimation, we develop a Hessian approximation method via parameter perturbation. This method captures second-order derivative information while maintaining the computational complexity of first-order gradients.
- We demonstrate that the uniformity of singular values in the extracted features is critical for model robustness. Based on this insight, we introduce a parameter robustness priori for $RP^2F$ to achieve a seamless posterior fusion.
- We conduct several experiments on the CIFAR-10, CIFAR-100, and Tiny-ImageNet datasets, demonstrating that $RP^2F$ supports backward knowledge transfer and achieves state-of-the-art performance.

## 2 RELATED WORKS

### 2.1 Incremental Learning

Incremental learning focuses on developing algorithms that can continuously learn from new data without forgetting previously acquired knowledge [12, 33]. Approaches within this field are broadly categorized into regularization-based methods, rehearsal-based methods, and dynamic architecture methods. In the following, we will review these three class methods.

**Regularization-based methods** integrate regularization loss terms to prevent forgetting. Some methods [1, 22, 29, 68] punish the changes in critical parameters containing old-task knowledge. Other knowledge-distillation-based methods hinge on the principle of transferring knowledge from an older model (teacher) to a newer model (student) during the IL process. A seminal work is Learning without Forgetting (LWF) proposed by Li and Hoiem [28], which utilizes knowledge distillation to preserve previously learned information while accommodating new knowledge. Based on LWF, several other notable methods [14, 26, 27, 47, 56, 57, 69, 71] have been developed. **Rehearsal-based IL methods** operate by retaining a subset of the original training data [3, 5, 6, 8, 9, 30, 38, 42] or generating synthetic data [49, 50, 65], which is then replayed with new data during subsequent training phases. Dark Experience Replay (DER) [7] combines experience replay with a distillation mechanism to maintain a balance between old and new knowledge. DER selectively stores a subset of the data from previous tasks and rehearses it alongside new data, thus preserving the model's performance on historical tasks. On the other hand, generative-replay-based methods leverage generative models, such as Generative Adversarial Networks (GANs) [16], to synthesize data from previous tasks, thereby eliminating the need to store real data. **Dynamic architecture methods** typically involve modifying the architecture of neural networks dynamically in response to new tasks. Some methods are based on masks [10, 19, 23, 34], such as PackNet [34]. PackNet prunes weights that are non-essential for learned tasks, thereby freeing up network capacity for new tasks, a process that strategically balances between retaining learned knowledge and adapting to new information. On the other hand, expansion-based methods [40, 53–55, 64] like Dynamically Expandable Networks (DEN) [66] selectively expand the network by adding neurons as needed.

As a subset of regularization-based methods, bayesian-based IL methods [20, 22, 25, 37, 39, 46, 61] utilise Bayesian inference through the learning process. Elastic Weight Consolidation (EWC) [22] applies a Bayesian perspective to determine the importance of neural network parameters, followed by imposing a constraint on significant deviations from previous parameters. IMM [25] proposes to merge the neural network parameters from previous tasks with the current task, which shares a similar motivation with the posterior fusion of $RP^2F$. Beyond IMM, $RP^2F$ further introduces parameter robustness priori and a more precise perturbation-based Hessian estimation method.

### 2.2 Robust Model Training

In the field of robust model training, the perturbation-based method is a primary research direction within the machine learning community. A common practice involves modeling parameter robustness using Lipschitz continuity and enhancing robustness by constraining the upper bound of the Lipschitz constant [2, 11, 18, 58]. However, recent studies [43] have highlighted that the Lipschitz constant is sensitive to the scale of model parameters and input features, and propose to employ condition numbers as a more precise metric of robustness. In the context of IL, robust methodologies have been integrated into some exemplar-based methods. They primarily concentrate on model robustness to exemplar variations. For instance, LiDER [5] forces the model to become robust to the changes of exemplars, thereby mitigating the over-fitting issue in exemplar-based methods. DRO [63] continually evolves the exemplar buffer to ensure the model learns more robust features. In contrast to these methods, our proposed $RP^2F$ does not rely on exemplar samples. Instead, we focus on investigating robustness against parameter variations, with the goal of mitigating information losses in posterior fusion, which contributes a novel perspective to IL.

## 3 PRELIMINARY

### 3.1 Problem Definition

Incremental learning (IL) is a cognitive process characterized by the gradual acquisition and assimilation of knowledge over time. As a popular setting of IL, Task Incremental Learning (Task-IL) [59] focuses on the sequential acquisition and retention of knowledge across multiple related tasks, where each task may involve distinct but related learning objectives.

Formally, let $D_1, D_2, \ldots, D_T$ denote datasets of $T$ classification tasks, where $D_t = \{(x_k^t, y_k^t \in Y_t)\}_{k=1}^{n_t}$ is associated with task $t$ that contains $n_t$ input-label pairs. These tasks feature disjoint classes, i.e., $Y_{t_1} \cap Y_{t_2} = \emptyset, \forall t_1 \neq t_2$. Task-IL sequentially presents these tasks to an IL model, with only the associated dataset $D_t$ being available at time step $t$. The model must adapt to task $t$ while also retaining knowledge of the former tasks. During the testing phase, the model's overall performance across all encountered tasks will be evaluated.

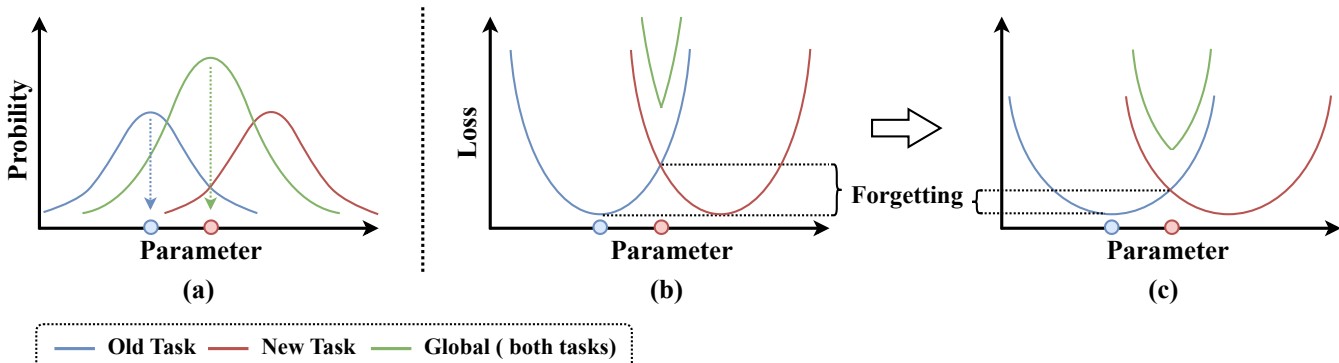

Figure 1: (a) RP$^2$F identifies the optimal parameter at the peak of the fusion posterior distribution. (b) After learning a new task, parameter adjustments shift from the peak posterior of the old task toward the maximum fusion posterior, leading to increased loss on the old task, i.e., forgetting. (c) The forgetting can be mitigated by enhancing parameter robustness against substantial changes.

## 3.2 Bayesian Incremental Learning

Incremental learning has been extensively explored within the Bayesian framework. The Bayesian IL framework leverages Bayes' rule to systematically update the posterior parameter distribution as new data becomes available. The fundamental concept lies in the Bayes' theorem, which provides an estimation of the parameter posterior as follows:

$$P(\theta|D) \propto P(D|\theta)P(\theta), \tag{1}$$

where $P(\theta|D)$ represents the parameter posterior after observing data $D$. This posterior is shaped by the parameter priori $P(\theta)$ and the likelihood $P(D|\theta)$.

In the context of IL, the posterior estimation after observing $t$ tasks in Eq. 1 can be extended into multiple tasks [20] as follows:

$$P(\theta|D_1, \ldots, D_t) \propto P(\theta) \prod_{t'=1}^{t} P(D_{t'}|\theta). \tag{2}$$

Using the posterior of all the former tasks as a priori, the posterior can be computed cumulatively:

$$P(\theta|D_1, \ldots, D_t) \propto P(\theta|D_1, \ldots, D_{t-1})P(D_t|\theta). \tag{3}$$

An integral part of Bayesian-based IL methods is the estimation of the posterior distributions $P(\theta|D_1, \ldots, D_{t-1})$, which is often computationally challenging for complex models. The Laplace approximation [31] offers a tractable solution by approximating the posterior distribution with a Gaussian centred at the maximum a posteriori (MAP) estimate. This approximation is particularly useful when the posterior is unimodal. The mathematical representation of this approximation is formalized as follows:

$$P(\theta|D) \approx \mathcal{N}(\theta|\mu_{MAP}, H_{MAP}^{-1}), \tag{4}$$

where $\mu_{MAP}$ is the mode of the posterior distribution (i.e., the MAP estimation), and $H_{MAP}^{-1}$ is the inverse of the Hessian around $\mu_{MAP}$, providing a second-order approximation of the posterior curvature.

Building upon this framework, bayesian-based IL methods [20, 22, 25] emerge to incorporate Bayesian principles into neural network training for each task. They typically mitigate catastrophic

forgetting by introducing a regularization term to the loss function that penalizes changes to important parameters:

$$\begin{aligned} \theta_{1:t} &= \arg\max_{\theta} \log P(\theta|D_1, \ldots, D_t) \\ &= \arg\max_{\theta} \log P(D_t|\theta) + \log P(\theta|D_1, \ldots, D_{t-1}) \\ &\approx \arg\min_{\theta} L_{ce}(\theta, D_t) + \frac{\lambda}{2}(\theta - \mu_{1:t-1})^{\top} \Lambda_{1:t-1}(\theta - \mu_{1:t-1}), \end{aligned} \tag{5}$$

where $L_{ce}$ represents the cross-entropy loss and $\lambda$ stands for a pre-defined hyperparameter weighting the regularizer. Here, the posterior $P(\theta|D_1, \ldots, D_{t-1})$ is approximated by the Gaussian distribution $\mathcal{N}(\theta|\mu_{1:t-1}, \Lambda_{1:t-1})$, where $\mu_{1:t-1}$ is typically assigned to the value of $\theta_{1:t-1}$ and $\Lambda_{1:t-1}$ signifies the importance matrix. The continuous update of $\Lambda_{1:t-1}$ incorporates the Hessian associated with the new task [22]:

$$\Lambda_{1:t} = \Lambda_{1:t-1} + H_t = \Lambda_{1:t-1} + \nabla_{\theta}^2 L_{ce}(\theta, D_t)|_{\theta=\theta_{1:t}}. \tag{6}$$

However, the computation of the Hessian matrix poses a significant challenge, due to its computational complexity (second-order partial derivatives) and the extensive storage requirements. In practice, a prevalent approach is to approximate the Hessian using the Fisher information matrix [20, 22, 39, 46]:

$$H_t \approx F_t = \nabla_{\theta} L_{ce}(\theta, D_t) \nabla_{\theta} L_{ce}(\theta, D_t)^{\top}|_{\theta=\theta_{1:t}}. \tag{7}$$

## 4 METHOD

### 4.1 Robust Bayesian IL Framework

Although Bayesian-based IL methods have demonstrated effectiveness in mitigating forgetting, they encounter significant challenges. One such challenge is **the control of regularization strength**. In Eq. (5), the cross-entropy loss and the regularization terms may not be scaled equivalently. This imbalance may lead $\theta_{1:t}$ fails to accurately represent the MAP estimation of posterior $P(\theta|D_1, \ldots, D_t)$. Manually adjusted balancing coefficients can not solve this issue, especially in the complex and unpredictable trajectory of neural network training. Additionally, **the issue of forgetting** arises as

parameters transition from the local optimum associated with previous tasks $\theta_{1:t-1}$ to a global optimum that encompasses new learning objectives $\theta_{1:t}$ (as illustrated in Figure 1 (b)). This forgetting becomes noteworthy when the performance is sensitive to parameter changes.

To address the aforementioned challenges, we propose the Robust Parameter Posterior Fusion (RP²F) framework for IL. Specifically, RP²F reformulates Eq. (2) by choosing a specific parameter robust priori $P(\theta)$ as follows:

$$P(\theta|D_1,\ldots,D_t) \propto \prod_{t'=1}^{t} P(\theta)^{\frac{1}{t}} P(D_{t'}|\theta). \quad (8)$$

Equation (8) enables us to model task-specific posterior solely based on the data from each respective task. In other words, instead of modeling the posterior of all encountered tasks $P(\theta|D_1,\ldots,D_{t-1})$ in Eq. (3), we are able to yield the MAP estimation directly for each individual task:

$$\theta_t^* = \arg\max_{\theta} \ \log P(\theta)^{\frac{1}{t}} P(D_t|\theta)$$

$$= \arg\max_{\theta} \ \log P(D_t|\theta) + \frac{1}{t} \log P(\theta) \quad (9)$$

$$\approx \arg\min_{\theta} \ L_{ce}(\theta, D_t) + \lambda L_{\text{robust\_pri}}(\theta),$$

where $L_{\text{robust\_pri}}(.)$ represents a parameter robustness priori weighted by $\lambda$. Notably, different from Eq. (5), Eq. (9) derives the MAP estimation only for the single task $t$.

Based on $\theta_t^*$, we apply the Laplace approximation to estimate $P(\theta)^{\frac{1}{t}} P(D_{t'}|\theta)$ with Gaussian $\mathcal{N}(\theta|\theta_t^*, H_t^{*-1})$, where $H_t^*$ represents the Hessian of the error around $\theta_t^*$, i.e., $H_t^* = \nabla_\theta^2 L_{ce}(\theta, D_t)|_{\theta=\theta_t^*}$. Taking into account the practical constraints of storage and computation, our approach aligns with prior works [22] that employ a diagonal matrix to estimate the Hessian matrix. The diagonal matrix, denoted by $\Lambda_t^*$, is thus regarded as indicative of the importance of the parameters. Building upon the Laplace approximation, we are able to derive the MAP estimation $\theta_{1:t}^*$ of the posterior distribution in Eq. (8):

$$\theta_{1:t}^* = \arg\max_{\theta} \ P(\theta|D_1,\ldots,D_t)$$

$$\approx \frac{\sum_{t'=1}^{t} \Lambda_{t'}^* \theta_{t'}^*}{\sum_{t'=1}^{t} \Lambda_{t'}^*}. \quad (10)$$

For the proof of Eq. (10), please refer to the supplementary materials.

Additionally, we present a task-accumulative version, designed to fulfil the requirements of IL settings:

$$\theta_{1:t}^* \approx \frac{\Lambda_{1:t-1}^* \theta_{1:t-1}^* + \Lambda_t^* \theta_t^*}{\Lambda_{1:t-1}^* + \Lambda_t^*}, \quad (11)$$

where $\Lambda_{1:t}^* = \Lambda_{1:t-1}^* + \Lambda_t^*$ also update in an accumulative manner.

Compared to the conventional Bayesian-based IL methods, RP²F has two specific designs. Firstly, RP²F determines the optimal parameter through the MAP estimate of the fusion parameter posterior for all encountered tasks in Eq (10). Unlike regularizer-based methods that struggle to balance the regularizer in Eq. (5), the posterior fusion manner facilitates a harmonious knowledge accumulation, achieving an enhanced equilibrium between maintaining previously learned information (stability) and incorporating new knowledge (plasticity). Secondly, the incorporation of the parameter-robustness priori mitigates potential forgetting induced by parameter updating.

To implement the RP²F framework, two key components remain to be devised: the efficient estimation of Hessian and the design of a parameters-robustness priori. Conventionally, the Hessian is estimated using the Fisher information matrix. Although the Fisher information matrix possesses many desirable properties [22], as described in Eq. (7), the common computation of Fisher inherently contains only first-order gradient information. This limitation is especially highlighted as previous works [22, 46] consider only the diagonal elements of the Fisher information matrix. To address this limitation, this paper introduces a novel Hessian matrix approximation method based on parameter perturbation, which effectively approximates second-order derivative information with the same computational complexity as first-order derivatives. In terms of the robustness priori to parameters, our investigation reveals that the uniformity in the eigenvalues of the features extracted by the network is closely linked to the robustness of the parameters. This observation motivates us to average the eigenvalues of features during training.

Next, Sections 4.2 and 4.3 elaborate on the Hessian approximation method and the parameter robustness priori, respectively.

## 4.2 Hessian Approximating via Parameter Perturbation

Previous works [22, 46] typically employ the diagonal Fisher information matrix to estimate the Hessian, focusing exclusively on the first-order derivatives as outlined in Eq. (7). In contrast, our research presents a novel Hessian estimation method based on parameter perturbation. The estimation is based on the gradient corresponding to infinitesimal parameter shifts, thereby providing a more precise and efficient approximation of Hessian.

To be more specific, considering the MAP estimation for $t$-th task posterior $\theta_t^*$, we introduce a minor perturbation $\delta$ to $\theta_t^*$. The second-order Taylor expansion on the empirical risk function is formulated as follows:

$$L_{ce}(\theta_t^* + \delta, D_t) \approx L_{ce}(\theta_t^*, D_t) + \nabla L_{ce}(\theta_t^*, D_t)^\top \delta + \frac{1}{2}\delta^\top H_t^* \delta, \quad (12)$$

where $H_t^* = \nabla_\theta^2 L_{ce}(\theta, D_t)|_{\theta=\theta_t^*}$ denotes the Hessian around $\theta_t^*$.

The expansion facilitates the analytical derivation of the model's sensitivity to parameter changes. Considering that $\theta_t^*$ is sufficiently trained such that the first-order gradient approaches zero, by setting $\delta = \eta \mathbf{1}$ where $\mathbf{1}$ signifies an all-one vector and $\eta$ represents an infinitesimally small constant, we differentiate both sides of Eq. (12) with respect to the perturbation $\delta$ yields the sum rows of the Hessian matrix:

$$\eta H_t^* \mathbf{1} \approx \left. \frac{\partial L_{ce}(\theta_t^* + \delta, D_t)}{\partial \delta} \right|_{\delta=\eta\mathbf{1}}. \quad (13)$$

By taking the assumption that the Hessian of a single-layer approaches diagonalization [44, 45], we estimate all the diagonal entries of $H_t^*$ with Eq. 13 layer by layer, ultimately forming the diagonal matrix $\Lambda_t^*$.

Estimating Hessian via perturbation significantly simplifies the complexity of Hessian computation. For a network with $n$ parameters, this method boasts a computational complexity of $O(n)$. In contrast, the direct computation of the Hessian, considering its need

to evaluate second-order derivatives for every pair of parameters, generally exhibits a computational complexity of $O(n^2)$, or even $O(n^3)$ when matrix inversion is involved. The efficient Hessian estimation enables a scalable and high-performance implementation of RP$^2$F.

## 4.3 Enhancing Parameter Robustness via Feature Regularization

In Section 4.1, by maximizing the fusion posterior, we derive the optimal estimation $\theta_{1:t}^*$. Intuitively, we aspire for $\theta_{1:t}^*$ to exhibit linear connectivity with the local optima for each task (e.g., $\theta_t^*$), aiming to minimize the difference $f(x; \theta_t^*) - f(x; \theta_{1:t}^*)$ for any $x \in D_t$. In other words, as depicted in Figure 1, our objective is to ensure that $\theta_t^*$ is robust to parameter changes, which motivates us to incorporate robustness priori into parameters during training.

The robustness of parameters is commonly characterized by the Lipschitz constant. However, the Lipschitz constant is sensitive to the scale of both the data and model parameters. As an alternative, inspired by [43], our work introduces a relative metric to assess the robustness of the model, which compares the rate of change in model outputs against the rate of parameter alterations:

$$R(\theta) = \sup_\delta \sum_{x \in D} \frac{\|f(x; \theta + \delta) - f(\theta)\| / \|f(x; \theta)\|}{\|\delta\| / \|\theta\|}, \quad (14)$$

where $\delta$ represents a perturbation applied to $\theta$. The numerator in (14) captures the relative change in model output due to parameter perturbation, while the denominator reflects the relative scale of the parameter perturbation. This formulation thus provides a measure of output sensitivity that is normalized by both the extent of the perturbation and the scale of the parameters, offering a more precise evaluation of the model's robustness.

Before delving into the robustness of deep neural models, we begin our examination with a simpler linear model $f(X; \Theta) = \Theta X$, where $X$ denotes the input feature matrix with each column corresponding to an individual input sample $x \in D$. For this linear model, the relative-robustness metric is bounded by:

$$\begin{aligned} R(\Theta) &= \sup_\Delta \frac{\|(\Theta + \Delta)X - \Theta X\| / \|\Theta X\|}{\|\Delta\| / \|\Theta\|} \\ &= \sup_\Delta \frac{\|\Delta X\|}{\|\Delta\|} \frac{\|\Theta\|}{\|\Theta X\|} = \|X\| \frac{\|\Theta\|}{\|\Theta X\|} \le \|X\| \|X^\dagger\|, \end{aligned} \quad (15)$$

where $X^\dagger$ represents the pseudo-inverse of $X$. When employing the Euclidean norm (also known as the $\ell_2$ norm) for the metric, $R(\Theta)$ is bounded by the ratio between the maximum singular value of $X$ to its minimum singular value:

$$R(\Theta) \le \frac{s_{max}(X)}{s_{min}(X)}. \quad (16)$$

Building upon the robustness analysis in the linear scenario, we extend our investigation to a deep neural network composed of $L$ layers [20]:

$$R(\theta) \le \prod_{l=1}^L R(\sigma^l) \prod_{l=1}^L R(\Theta^l) \le M \prod_{l=1}^L \frac{s_{max}(X^l)}{s_{min}(X^l)}, \quad (17)$$

where $X^l$, $\Theta^l$, and $\sigma^l$ denote the input features, parameters, and activation units of the $l$-th layer, respectively. We suppose that the

robustness of all activation units is bounded by a positive constant $M$, i.e., $\prod_{l=1}^L R(\sigma^l) \le M$.

Equation (17) elucidates that the robustness of the network parameters is governed by the uniformity of the singular values of the input features at each layer (extracted by the preceding layer). Yujun et al. [48] have demonstrated that this uniformity can be enhanced by penalizing the Frobenius norm of the covariance matrix. Based on this insight, we introduce the following priori to enhance the robustness of network parameters:

$$L_{\text{robust\_pri}}(\theta) = \frac{1}{L} \sum_{l=1}^L \|\hat{X}^{l\top} \hat{X}^l\|_F^2, \quad (18)$$

where $\hat{X}^l$ represents the normalized feature matrix of the $l$-th layer. However, Eq. (18) presents a considerable challenge in terms of computational complexity, particularly for high-dimensional deep neural networks. Given that features in the shallow layers of neural networks tend to approach a full rank [15], in practice, we strategically apply regularization only to the features extracted from the penultimate layer.

## 4.4 Incremental Learning with Robust Parameter Posterior Fusion (RP$^2$F)

This section elaborates on our IL algorithm, which leverages the robust parameter posterior fusion framework. This approach integrates the perturbation-based Hessian approximation and the parameter-robustness priori proposed in Section 4.2 and 4.3, respectively. We will describe the training procedure of our method on a multi-head model consisting of a feature extractor $\theta$ and $T$ task-specific classifier $\tau_1, \ldots, \tau_T$.

**Learning task 1.** During the phase of Task 1, the feature extractor alongside the classifier undergoes joint training using dataset $D_1$. The training leverages cross-entropy loss and parameter robustness priori to enhance performance and ensure robustness:

$$\theta_1^*, \tau_1^* \longleftarrow \underset{\theta_1, \tau_1}{\arg\min} \, L_{ce}(\tau_1 \circ \theta_1, D_1) + \lambda L_{\text{robust\_pri}}(\tau_1 \circ \theta_1, D_1), \quad (19)$$

where $\tau \circ \theta$ denotes a unified network that the classifier $\tau$ operating on features extracted by the feature extractor $\theta$. Upon the completion of training for task 1, we assign the parameters of an additional fused posterior extractor with $\theta_{1:1}^* \longleftarrow \theta_1^*$, and approximate the Hessian via $\Lambda_{1:1}^* \longleftarrow \frac{\partial L_{ce}(\theta_1^* + \delta, D_t)}{\eta \partial \delta}$.

**Learning the following tasks ($t$ for illustration).** We initialize $\theta_t$ with the last optimal parameters $\theta_{t-1}^*$ learned from the previous task. This strategy aims to keep the optimization process for all tasks within the same vicinity of a local optimum, thereby facilitating effective fusion of the posterior distribution [25, 52]. Subsequently, during **each training epoch**, the model undergoes a four-step updating process: 1) training the feature extractor $\theta_t$; 2) estimating the Hessian $\Lambda_t$ surrounding $\theta_t$; 3) updating the fused posterior extractor $\theta_1 : t$; 4) training the task-specific classifier $\tau_t$.

The formulations for these four steps are presented as follows:

$$\theta_t \longleftarrow \theta_t - \alpha \nabla_{\theta_t} L_{ce}(\tau_t \circ \theta_t, D_1) + \lambda L_{\text{robust\_pri}}(\tau_t \circ \theta_t, D_t)$$

$$\Lambda_t \longleftarrow \frac{\partial L_{ce}(\theta_t + \delta, D_t)}{\eta \partial \delta}$$

$$\theta_{1:t} \longleftarrow \frac{\Lambda_{1:t-1}^* \theta_{1:t-1}^* + \Lambda_t \theta_t}{\Lambda_{1:t-1}^* + \Lambda_t} \tag{20}$$

$$\tau_t \longleftarrow \tau_t - \alpha \nabla_{\tau_t} L_{ce}(\tau_t \circ \theta_{1:t}, D_1),$$

where $\alpha$ denotes the learning rate. This iterative update is designed to ensure that the classifier $\tau_t$ remains compatible with the optimal fused posterior extractor $\theta_{1:t}$. Upon the training convergence, we obtain the optimal values of $\theta_t^*$, $\Lambda_t^*$, $\theta_{1:t}^*$, and $\tau_t^*$. Among them, $\Lambda_t^*$ is utilized to refine the variance of the fused posterior $\Lambda_{1:t}^* \longleftarrow \Lambda_{1:t-1}^* + \Lambda_t^*$. $\Lambda_{1:t}^*$, along with the others, is maintained for the subsequent task training.

**Inference.** During the testing phase, given sample $x$ associated with task identification $t$, we select the corresponding classifier $\tau_t^*$ along with fused extractor $\theta_{1:T}^*$ to perform a prediction:

$$\hat{y} = f(x; \tau_t^* \circ \theta_{1:T}^*), \tag{21}$$

where $\hat{y}$ denotes the classification result.

## 5 EXPERIMENTS

### 5.1 Settings

**Datasets.** To ensure a comprehensive evaluation, we select the following three datasets to conduct experiments:

- **5-split CIFAR-10 [24].** The CIFAR-10 dataset consists of 60,000 color images of $32 \times 32$ pixels distributed across 10 different classes, with each class containing 6,000 images. In our experiments, CIFAR-10 is divided into 5 splits based on the class labels.
- **10-split CIFAR-100 [24].** Similar to CIFAR-10, CIFAR-100 features 100 classes containing 600 images each. We partition CIFAR-100 into 10 splits, with each containing 10 classes.
- **10-split Tiny-ImageNet [51].** Tiny-ImageNet is a scaled-down version of the ImageNet dataset, consisting of 200 classes, each with 600 $64 \times 64$ color images. For our study, this dataset is segmented into 10 splits, each with 20 classes.

**Performance metrics.** To thoroughly evaluate the performance of $RP^2F$, we employ two metrics [7, 30]: classification accuracy (ACC) and backward knowledge transfer (BWT). ACC is calculated after the model has been sequentially trained on all tasks. This metric serves as a direct indicator of the model's ability to classify images correctly across all classes, reflecting its overall performance in IL. BWT is designed to quantify how learning new tasks affects the performance of previously learned tasks. A negative value of BWT indicates forgetting, while a positive value suggests that learning new tasks has beneficial effects on previous tasks' performance. The formula for calculating BWT is defined as follows:

$$BWT = \frac{1}{T-1} \sum_{t'=1}^{T-1} (ACC_{T,t'} - ACC_{t',t'}), \tag{22}$$

where $ACC_{j,i}$ represents the accuracy on task $i$ once the model finish learning task $j$. To ensure the reliability of results, we conduct all experiments five times with random seeds and report the average performance.

**Baselines.** We compare our method with various latest and classic IL methods, including Learning without Forgetting (LwF) [28], Synaptic Intelligence (SI) [68], Gradient Episodic Memory (GEM) [30], online Elastic Weight Consolidation (oEWC) [46], Learning without Memorizing (LwM) [13], Dark Experience Replay (DER and DER++) [7], Efficient Feature Transformation (EFT) [60], Pototype Augmentation and Self-Supervision (PASS) [70], Gradient Projection Memory (GPM) [41], Adam-NSCL [62], Always Be Dreaming (ABD) [50], Complementary Learning System (CLS-ER) [3], Filter Atom Swapping (FAS) [36], DCPOC [53], PRAKA [47], and MIND [4]. We also report the performance of a base model (referred to as Joint), which is trained jointly using data from all tasks. Clearly, Joint does adhere to the Task-IL setting, and its results are typically considered as the upper bound for IL methods. Additionally, for methods specifically designed for Class-IL, we employ a multi-head version of them to obtain the experimental results.

**Implementation details.** The experiments are conducted following the requirements of the Task-IL setting, where only the corresponding dataset is available for training in each task. For exemplar-based methods, we provide them with an extra sample buffer that can store up to 500 samples. We optimize the model parameters using Stochastic Gradient Descent (SGD), with learning rates adjusted specifically for each dataset: 0.2 for CIFAR-10, 0.05 for CIFAR-100, and 0.3 for Tiny-ImageNet. The hyperparameter $\lambda$ (weighting the parameter-robustness regularize in Eq. (9)) is set to 1e-6 for CIFAR-10 and 1e-5 for both CIFAR-100 and Tiny-ImageNet. Further details are available in our supplementary code.

### 5.2 Comparison Result

In this section, we present a comprehensive comparison of our proposed $RP^2F$ method against several state-of-the-art baselines as detailed in Section 5.1. Table 1 presents the average accuracy (%) over five runs on 5-split CIFAR-10, 10-split CIFAR-100, and 10-split Tiny-ImageNet.

The Joint model, which serves as the theoretical upper bound performance, consistently shows the highest accuracy across all datasets. We also found that some replay-based methods, such as GEM, DER, DER++, and ABD, exhibit competitive performance on small-scaled CIFAR-10, but experience a decline in efficacy on the CIFAR-100 and Tiny-ImageNet datasets. This observation suggests that the replay buffers and generative models may be insufficient for modeling complex data distributions.

Additionally, EFT achieves the highest accuracy on the CIFAR-10 dataset with 95.09%, surpassing $RP^2F$. The observed discrepancy in the performance of $RP^2F$ may be attributed to the inherent structure of CIFAR-10's 5-split configuration, where each task only contains data of two classes. This limited class diversity may lead to overfitted parameters, which further affect the accurate estimation of posterior within $RP^2F$. Nevertheless, $RP^2F$ outperforms all other baselines on the more challenging CIFAR-100 and Tiny-ImageNet datasets, suggesting its superior capability to handle more diverse and complex datasets.

Table 1: Comparison results on several datasets. We report the average accuracy (%) over five runs with random seeds, and the higher the better. (*) indicates the upper-bound model that is jointly trained with all tasks.

| Methods | Venue | CIFAR-10 | CIFAR-100 | Tiny-ImageNet | Average |
|---|---|---|---|---|---|
| Joint* | - | 98.07 | 91.18 | 82.01 | 90.42 |
| LwF [28] | TPAMI2017 | 91.91±0.7 | 63.78±4.3 | 58.61±1.8 | 71.43 |
| SI [68] | ICML2017 | 76.15±2.6 | 62.21±2.6 | 60.91±1.3 | 66.42 |
| GEM [30] | NIPS2017 | 85.14±2.1 | 62.80±2.7 | 44.66±1.7 | 64.20 |
| oEWC [46] | ICML2018 | 64.17±4.8 | 38.40±1.9 | 31.91±0.9 | 44.83 |
| LwM [13] | CVPR2019 | 78.01±0.8 | 68.88±0.9 | 45.57±0.2 | 64.15 |
| DER [7] | NIPS2020 | 93.13±0.3 | 73.26±1.3 | 51.22±1.5 | 72.54 |
| DER++ [7] | NIPS2020 | 93.71±0.4 | 74.86±1.1 | 53.00±0.4 | 73.86 |
| EFT [60] | CVPR2021 | 95.09±0.3 | 79.29±0.6 | 63.88±0.5 | 79.42 |
| PASS [70] | CVPR2021 | 86.07±0.2 | 77.30±0.4 | 62.87±0.4 | 75.41 |
| GPM [41] | ICLR2021 | 86.58±0.9 | 70.93±0.9 | 59.84±0.2 | 72.45 |
| Adam-NSCL [62] | CVPR2021 | 87.23±0.4 | 65.69±0.2 | 59.98±0.7 | 70.97 |
| ABD [50] | ICCV2021 | 95.11±0.3 | 74.83±0.5 | 46.76±0.6 | 72.23 |
| CLS-ER [3] | ICLR2022 | 93.53±0.3 | 72.11±0.5 | 57.36±0.7 | 74.33 |
| FAS [36] | ICLR2022 | 90.89±1.3 | 70.89±0.6 | 60.10±0.2 | 73.96 |
| DCPOC [53] | PR2023 | 90.43±0.3 | 72.20±0.2 | 53.08±0.2 | 71.90 |
| PRAKA [47] | ICCV2023 | 83.74±0.5 | 76.21±0.4 | 63.50±0.3 | 74.48 |
| MIND [4] | AAAI2024 | **95.67±0.7** | 77.33±0.5 | 63.82±0.5 | 78.94 |
| RP$^2$F (ours) | - | 91.65±0.3 | **83.06±0.2** | **65.81±0.6** | **80.17** |

Table 2: Ablation experiment results (ACC %) of RP$^2$F on CIFAR-100 and Tiny-ImageNet.

| Hessian estimation | robustness priori | 10-split CIFAR-100 | 10-split Tiny-ImageNet |
|---|---|---|---|
| Identity matrix | × | 79.42±0.66 | 63.96±0.30 |
| Identity matrix | ✓ | 79.91±0.28 | 64.62±0.38 |
| Fisher information matrix | × | 82.29±0.16 | 64.40±0.57 |
| Fisher information matrix | ✓ | 82.85±0.36 | 64.79±0.47 |
| Parameter-perturbation based (ours) | × | 82.37±0.49 | 64.78±0.70 |
| Parameter-perturbation based (ours) | ✓ | **83.06±0.2** | **65.81±0.6** |

## 5.3 Ablation Study

This section delves into the ablation study to analyse the effectiveness of the parameter-perturbation-based Hessian estimation method and the parameter-robustness priori. Specifically, we explore three Hessian estimation strategies: the identity matrix, the Fisher information matrix, and the proposed parameter-perturbation-based approach. For each strategy, we further investigate the impact of introducing the parameter-robustness priori. The results on CIFAR-100 and Tiny-ImageNet are presented in Table 2.

We begin with the identity matrix as the simplest form of Hessian approximation, which yields an accuracy of 79.42% on CIFAR-100 and 63.96% on Tiny-ImageNet. Equipping the parameter-robustness priori provides a slight enhancement, improving accuracies to 79.91% and 64.62%, respectively. Shifting to the widely used Fisher information matrix, we observe a notable improvement in performance. Without the regularizer, it achieves 82.29% accuracy on CIFAR-100 and 64.40% on Tiny-ImageNet. Integrating the parameter-robustness priori further boosts the performance to 82.85% and

64.79%, respectively. Our novel parameter-perturbation-based Hessian estimation method is more competitive than the Fisher information matrix. Without the regularizer, it attains an accuracy of 82.37% on CIFAR-100 and 64.78% on Tiny-ImageNet. The incorporation of the parameter-robustness priori pushes the performance to the highest accuracy: 83.06% on CIFAR-100 and 65.81% on Tiny-ImageNet.

All the results underscore the efficacy of parameter-perturbation-based Hessian estimation in accurately reflecting the true landscape of the loss function. Furthermore, these findings emphasize the ability of the parameter-robustness priori to enhance the model's robustness and mitigate forgetting.

## 5.4 Analysis of Backward Knowledge Transfer

This section analyzes the backward knowledge transfer of RP$^2$F, a critical IL ability to retain or even enhance performance on previously learned tasks when new tasks are introduced. Our discussion is based on empirical evidence in Figure 2 and Table 3.

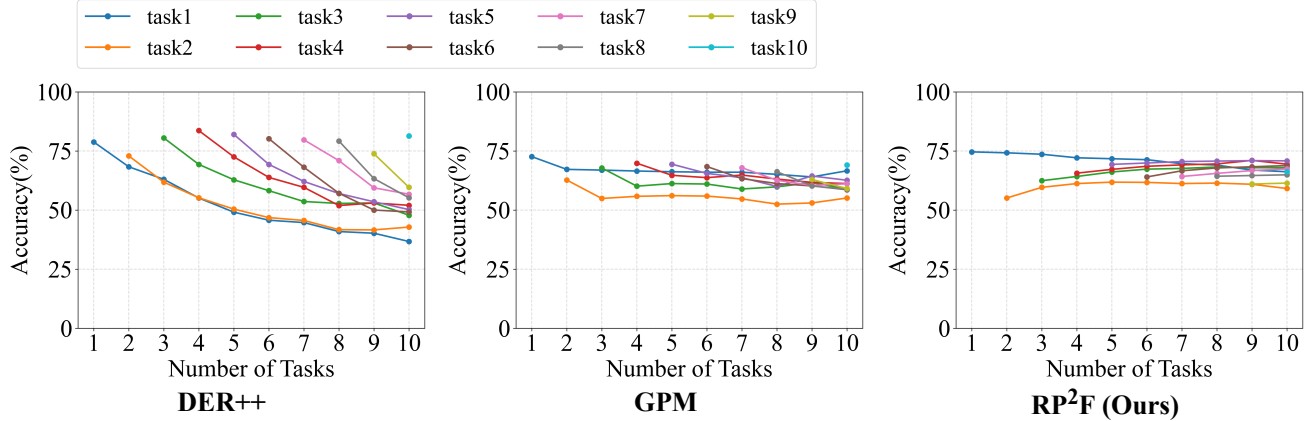

Figure 2: Accuracy trends over tasks on Tiny-ImageNet for DER++ (left), GPM (middle), and our proposed RP$^2$F model (right).

Table 3: Backward knowledge transfer (BWT, %) on CIFAR-100 and Tiny-ImageNet.

| Methods | DER++ [7] | LWF [28] | OWM [67] | PASS [70] | GPM [41] | Adam-NSCL [62] | PRAKA [47] | RP$^2$F (ours) |
|---|---|---|---|---|---|---|---|---|
| CIFAR-100 | -12.56±5.7 | -20.91±3.6 | -11.25±0.9 | -9.34±0.7 | -3.52±0.3 | -2.24±0.3 | -4.09±0.3 | **0.56±0.2** |
| Tiny-ImageNet | -29.06±1.2 | -23.84±1.6 | -3.30±0.2 | -3.23±0.4 | -7.91±0.5 | -5.58±0.6 | -5.42±0.1 | **0.92±0.8** |

Figure 2 visually illustrates the performance trajectory of each task through IL. As can be seen, DER++ exhibits a declining trend in accuracy for earlier learned tasks, signifying the occurrence of forgetting. In contrast, GPM maintains a more stable performance throughout the learning phases, showing its effectiveness in mitigating forgetting. Remarkably, RP$^2$F exhibits improved performance on older tasks with the introduction of new tasks (with the exception of the first task).

Further quantitative evidence of BWT is presented in Table 3. The BWT of several methods are calculated on CIFAR-100 and Tiny-ImageNet, including DER++, LwF, OWM, PASS, GPM, Adam-NSCL, and RP$^2$F. It is evident that all baseline methods exhibit a negative BWT, highlighting the challenge of forgetting in IL. In contrast, RP$^2$F records a positive BWT of 0.56% on CIFAR-100 and 0.92% on Tiny-ImageNet, surpassing all other methods. Both visual illustration and quantitative values of BWT underscore RP$^2$F's superior ability to backward knowledge transfer.

## 5.5 Sensitivity Analysis of $\lambda$

This section provides a sensitivity analysis of the hyper-parameter $\lambda$, responsible for weighting the parameter-robustness priori in Eq. (9). We conduct experiments with various values of $\lambda$ and summarize the results in Figure 3. As can be seen, setting $\lambda$ to zero or an excessively high value results in suboptimal performance of RP$^2$F, attributing to the ineffectiveness or overwhelming impact of the regularizer, which in turn perturbs the training dynamics. Nevertheless, a stable performance window for $\lambda$ is observed within the range of $[1e-7, 1e-5]$. Based on these empirical findings, we recommend setting $\lambda$ to $1e-5$ on CIFAR-100 and Tiny-ImageNet to achieve the best overall performance.

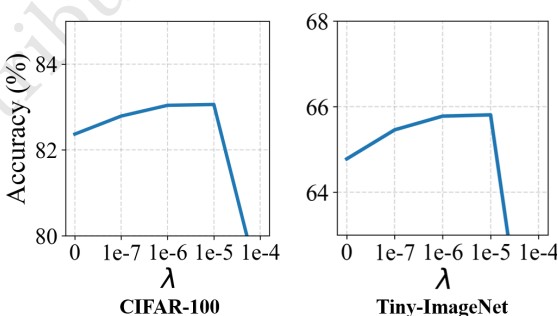

Figure 3: Accuracy of RP$^2$F with various values of $\lambda$ (Cf. Eq.(9)) on CIFAR-100 (left) and Tiny-ImageNet (right).

## 6 CONCLUSION

In this work, we propose the Robust Parameter Posterior Fusion (RP$^2$F) framework for incremental learning. RP$^2$F models and fuses the parameter posterior distributions from both new and existing tasks, facilitating a more equitable integration of knowledge across tasks. Furthermore, we incorporate robustness priori into RP$^2$F to mitigate potential forgetting induced by posterior fusion. By employing MAP estimation on the fused posterior, RP$^2$F achieves a harmonious knowledge integration.

RP$^2$F has certain limitations, such as relying on task boundaries to perform posterior fusion. In future work, we plan to address this issue by incorporating online learning techniques to extend the applicability of our method.

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
