# OpenReview forum: "Incremental Learning via Robust Parameter Posterior Fusion"
_acmmm.org/ACMMM/2024/Conference — MM2024 Poster_

### Official Review · Reviewer_HGCW · 2024-05-24

**Rating:** 4
**Confidence:** 3

**Summary:**

This paper proposes a novel Bayesian incremental learning framework called Robust Parameter Posterior Fusion (RP2F), which effectively alleviates the catastrophic forgetting problem and realizes reverse knowledge transfer by directly estimating and fusing the parameter posteriors of new and old data.

**Strengths:**

1. To my best knowledge, this approach is novel. The authors clearly derive the methodology in the paper, which I think is sound.

2. The comparison method is sufficient, and the proposed method performs well on key incremental learning tasks

3. The background information is sufficient and clear.

**Limitations:**

1. I am interested in validating the proposed method on tasks other than classification.

2. In sec 4.2, Eq. (13), the derivation of the approximation of the Hessian may be wrong. If I understand correctly, this paper uses the finite difference method and assumes that the gradient of the trained parameters is 0, then the numerator in the formula should be changed to $\delta$. Eq. represents the gradient of $\delta$ and cannot approximate the Hessian.

**Suitability:**

2

---

### Official Review · Reviewer_ZFPT · 2024-05-24

**Rating:** 4
**Confidence:** 2

**Summary:**

The paper presents the Robust Parameter Posterior Fusion (RP2F) framework, which directly estimates the parameter posterior for new data without introducing extra loss regularization. This method effectively mitigates catastrophic forgetting and achieves backward knowledge transfer.

**Strengths:**

(1) Well-motivated approach to improve Incremental Learning;

(2) The proposed approach achieves great results compared with sufficient methods.

**Limitations:**

(1) Eq. (13) uses the finite difference method to derivate the approximation of the Hessian under the assumption that the gradient of the trained parameters is 0. However, the result is incorrectly expressed as the gradient with respect to $\delta$, which should be modified to $\frac{\partial L_{ce}\left(\theta _{t}^{*}+\delta ,D_t \right)}{\delta}$.

(2) A comparison of computational efficiency and memory usage between the proposed method and existing methods should be provided.

**Suitability:**

2

---

### Official Review · Reviewer_6VCT · 2024-05-28

**Rating:** 5
**Confidence:** 3

**Summary:**

This paper presents a method for incremental learning called as Robust Parameter Posterior Fusion. In this approach, the posterior on the backbone of the feature extracted for each task is learned separately and then fused together based on MAP principle. It also includes a shared parameter priori-based regularization to ensure robustness of the feature extractor parameters. Through extensive empirical experiments the authors demonstrate the promising results of their method when compared against prominent baselines in incremental learning.

**Strengths:**

- The paper presents a very novel approach for incremental learning (such as the inclusion of Hessian approximation using parameter perturbation), and technically it seems very strong & practical based on the assumptions and the design choices made. I expect this method to be valuable to the practitioners of this field.

- The experiments are exhaustive, and the metrics are also very relevant. The results are very promising overall, with the proposed method outperforming on overall accuracy when compared against all the baselines on CIFAR-100 & Tiny-ImageNet.

- The approach is well described, especially when one thinks from the perspective of implementation. Also, the authors have done a good job at describing the prior work in this area.

**Limitations:**

- An important piece of this method seems to be the parameter-robustness priori. However, based on the current presentation, it's very hard to wrap my head around the significance of this piece (especially from an intuition perspective). I'd encourage the authors to provide an intuition behind what this does, and why is this important.  Furthermore, it seems that we still need to tune an hyperparameter $\lambda$ as we did in eq (5). Could you please describe the difference with current method and eq(5) in more detail so that it's easy to understand?

- Could you please describe some applications where this method would be applicable, and which would be of the interest to ACM MM audience?

- A small suggestion: Having an algorithm block would be nice.

- Line 474-475: "..to exhibit the linear connectivity with local optima for each task.." - why is this the case? can you please explain?

- A high level question - how do we think about meta learning in the context of incremental learning? I understand that in meta learning, at test time we would have to adapt to a new task, but can we think about the meta parameters as the latent representation of the posterior over all tasks?

**Suitability:**

2

---

### Meta-Review · Area_Chair_P8Wn · 2024-07-02

**Recommendation:** Accept (Poster)
**Confidence:** 5

**Metareview:**

The reviewers agree that the submission involves a novel approach. 6VCT calls the experimental approach “exhaustive” and the results “very promising”. ZFPT adds that the approach to incremental learning is “well-motivated” and HGCW additionally felt the background was “sufficient and clear”.

In terms of limitations 6VCT wanted further information about the parameter-robustness priori which was provided in the rebuttal. ZFPT felt a comparison of time/space efficiency was warranted. HGCW wanted to see tasks other than classification.

Two of the reviewers felt that concerns were sufficiently addressed in the rebuttal.

All reviewers agreed that this paper is worthy of aceptance (1 weak, 2 borderline). Given the arguments that this work is novel, achieves good results, and is of interest to the MM community I recommend that it is accepted with updates as noted in the rebuttal. This recommendation is only tempered by the reviewer’s rating that this is “moderately suitable” work, therefore perhaps more appropriate as a poster.